# YEAST: Yet Another Sequential Test

**Alexey Kurennoy**
Meta
akurennoy@meta.com

**Majed Dodin**
Delivery Hero
majed.dodin@deliveryhero.com

**Tural Gurbanov**
SoundCloud
tural.gurbanov@soundcloud.com

**Ana Peleteiro Ramallo**
Preply
ana.peleteiro@preply.com

## Abstract

The online evaluation of machine learning models is typically conducted through A/B experiments. Sequential statistical tests are valuable tools for analysing these experiments, as they enable researchers to stop data collection early without increasing the risk of false discoveries. However, existing sequential tests either limit the number of interim analyses or suffer from low statistical power. In this paper, we introduce a novel sequential test designed for the continuous monitoring of A/B experiments. We validate our method using semi-synthetic simulations and demonstrate that it outperforms current state-of-the-art sequential testing approaches. Our method is derived using a new technique that "inverts" a bound on the probability of threshold crossing, based on a classical maximal inequality.

## 1 Introduction

Online A/B experiments have become a standard approach for evaluating the impact of machine learning models on the systems and products where they are deployed. Although this approach offers significant advantages, it also poses the risk of exposing users to harmful changes that could degrade their experience. Unlike traditional techniques such as the Student's t test, sequential testing supports repeated significance checks without increasing the false positive rate. This makes it well-suited for real-time monitoring of experiments, allowing early detection of harmful effects, which helps protect user experience and reduce financial losses.

Sequential testing methods can be broadly divided into discrete and continuous. Discrete sequential tests permit only a fixed number of interim checks while continuous methods impose no such limit and allow checking for significance at the experimenter's will - even after each new observation. In this paper, we focus on continuous sequential tests. We find that the absence of a restriction on the number of interim checks is useful in practice, as it avoids a situation where the experimenter and the stakeholders observe a degradation in the metric of interest but have to wait until the next checkpoint before assessing the statistical significance and making decisions.

In this paper, we propose a novel continuous sequential test that demonstrates a better control of Type-I error and a greater power than current state-of-the-art methods for continuous experiment monitoring. The theoretical technique we use to derive and (asymptotically) justify this novel test is, to the best of our knowledge, also new.

We validate the proposed sequential test using a semi-synthetic simulation experiment based on a public real-world data set. We share the associated code for greater reproducibility [16].

To summarise, our **main contributions** are as follows.

39th Conference on Neural Information Processing Systems (NeurIPS 2025).

- We propose a novel (asymptotic) sequential test. This test supports unlimited interim significance checks.

- We present a new theoretical technique that consists of inverting the bounds obtained from the (generalised) Levy inequalities [21, Paragraph 29.1, A].

- We conduct an empirical study and demonstrate that the proposed sequential test has better Type-I error control and higher power than current state-of-the-art methods for continuous experiment monitoring.

- We share the associated code [16] for greater reproducibility and to promote further research on this topic.

The remainder of this paper is structured as follows. Section 2 reviews related work. Section 3 introduces the proposed method, explains its derivation, and then presents its formal theory. In Section 4, we assess the correctness of the proposed method using a semi-synthetic experiment (based on a public real-world data set). Section 5 discusses the limitations of the proposed approach and Section 6 concludes.

## 2    Related Work

Our work is related to a large literature on sequential decision-making based on sequences of hypothesis tests, confidence sequences, or sequences of "always-valid" p-values. While early work in this field was primarily concerned with clinical trials and industrial quality control [6, 4, 2], more recent work is typically motivated by the sequential nature of online experiments, where users arrive sequentially and generate outcomes that are observed quickly relative to the duration of the experiment [17, 14, 11]. Here, the goal of sequential testing is typically either to reduce the opportunity cost of longer experiments, as in the context of best-arm identification in multi-armed bandit problems [13, 15, 39, 12], or to manage risk by faster detection of harmful treatments [20, 10].

### 2.1    Martingale Methods and E-Processes

Building on the seminal work of Wald [36] and Doob [7], most modern sequential procedures are based on maximal inequalities for appropriately constructed martingales that provide time-uniform statistical guarantees. While Wald's original sequential probability ratio test (SPRT) was developed from scratch, its construction is essentially of this "martingale-type". SPRT sequentially evaluates the likelihood ratio test statistic at each sample size for the null hypothesis $H_0 : \theta = \theta_0$ against the simple alternative $H_1 : \theta = \theta_1$. The test continues until the statistic crosses one of two predefined bounds, at which point the null hypothesis is either rejected or not rejected accordingly. SPRT requires specifying a single-point alternative, which limits its use for general-purpose experimental monitoring. To address this limitation, likelihood ratio methods using the method of mixtures (mSPRT) have been developed. These approaches replace the single-point alternative with a composite hypothesis and the standard likelihood ratio — with a mixture, which remains a martingale under the null [27].

In addition to mSPRT, the class of martingale methods encompasses more sophisticated approaches constructed using other martingales and falling under the framework of generalized always-valid inference (GAVI) [14, 11, 10, 19, 18, 3].

Another recent branch of martingale-based methods is built upon the principle of testing by betting (see [25] and references therein).

The martingale-based approach was generalised through the concept of *e-processes* [9, 30] — stochastic processes characterised by a property that allows for straightforward construction of sequential tests. The roots of this perspective can be traced back to [35], who introduced the use of test martingales within a game-theoretic framework as a foundation for statistical testing. This remains an active area of research, with a growing literature developing new constructions and applications of e-processes [31, 32]. See [26] for a recent overview of the topic and further references.

### 2.2    Group Sequential Methods

A conceptually different approach is followed by methods based on group sequential tests (GST) [24, 23, 33, 8]. GST methods rely on (asymptotic) characterizations of the joint distribution of

a vector of test statistics across increments of data, which permits the computation of sequential boundaries that uniformly control the false discovery rate via multivariate (numerical) integration. Earlier GST methods (such as [24, 23]) required a fixed number and timing of interim analyses. In contrast, the later generalisation [8] permits flexible (in theory up to continuous) monitoring. This flexibility is achieved using an alpha-spending function, which controls the allocation of Type I error across analyses, based on a prespecified maximum sample size.

## 2.3 Simple Sequential A/B Testing

Our approach has similarities to *simple sequential A/B testing* [22]. However, the test from [22] only allows monitoring count-type metrics (for example, the number of clicks or purchases), whilst our proposed method can be used to monitor metrics of any type, including real-valued financial metrics such as revenue. One can view our work as a generalisation of [22], yet we emphasise that this generalisation is not at all trivial as the theory presented in [22] fundamentally supports only count-type metrics.

# 3 YEAST: YEt Another Sequential Test

In this section, we present a novel, alternative sequential testing method proposed in this paper.

## 3.1 Problem Setup and Notation

Suppose we are conducting an A/B test and have a certain metric of interest. We want to constantly track this metric and have a tool to check if the difference between the groups is at any point large enough to conclude that the treatment has an impact on the metric. The tool must have good control of the false detection rate despite the repeated checks. In other words, it must allow for peeking into the interim results of the experiment without inflating the false detection rate.

To develop a tool satisfying the above requirement, we consider the difference $S$ in the total value of the metric of interest accumulated by the control ($W = 0$) and the treatment ($W = 1$) groups. Typically, this difference $S$ can be represented as a sum of increments over a stream of events. For example, if the metric of interest is revenue then the tracked difference $S$ can be decomposed into the sum of order values over checkout events where the order value is taken with a positive sign if it comes from the control group and with a negative sign if it comes from the treatment arm. Mathematically, the increments can be written as

$$X_i = (1 - 2W_i)\, Y_i, \tag{1}$$

where $Y_i$ is the value or outcome associated with the $i$-th event (such as the order value in the example above) and $W_i$ is the event assignment indicator (it equals 0 if the event was generated by a subject from the control group and 1 if its subject belongs to the treatment group). With this notation at hand, the difference in the metric of interest after observing $n$ events can be written as the running sum

$$S_n = \sum_{i=1}^{n} X_i, \quad n \geq 1, \tag{2}$$

as illustrated in the example data set displayed in Table 1.

Table 1: An Example of Experiment Data

| i | timestamp | group | Y | X | S |
|---|---|---|---|---|---|
| 1 | 2023-08-01 12:00:00 | control | 175.0 | 175.0 | 175.0 |
| 2 | 2023-08-01 12:00:02 | treatment | 35.5 | -35.5 | 139.5 |
| 3 | 2023-08-01 12:00:05 | treatment | 20.0 | -20.0 | 119.5 |
| 4 | 2023-08-01 12:00:10 | control | 100.0 | 100.0 | 219.5 |
| ... | ... | ... | ... | ... | ... |

So far we have used revenue monitoring as an example but one can think of examples from other domains such as *video streaming* (metric: hours streamed; the events are user sessions; the event value

is the total duration of videos streamed within the session), *online advertisement* (metric: number of clicks; the events are clicks; the event value is 1), *medical drug trial* (metric: number of cases of side-effects; the events are incoming patients; the event value is the indicator of side-effect presence after taking the drug), *online subscription services* (metric: number of new users who convert into a paid subscription; the events are newly acquired users; the event value is the indicator of subscribing), *banking* (metric: total transaction fee; the events are transactions; the event value is fee amount), etc.

## 3.2 Assumptions and the Null Hypothesis

We will now state and discuss our assumptions and the null hypothesis.

**Assumption 1 (Random Subject Assignment)** *The subjects are assigned to the control or treatment uniformly at random and with equal probability.*

We work with the following null hypothesis.

$\mathbf{H}_0$: *the treatment does not affect either the event outcome distribution or the frequency of events, so that*

(a) the event outcomes and the assignment indicators are independent from each other, i.e.,

$$\{Y_1, Y_2, \ldots\} \perp\!\!\!\perp \{W_1, W_2, \ldots\}, \tag{3}$$

(b) and the frequency of events in each group is the same, implying (under Assumption 1) that

$$\Pr\{W_i = 1\} = 0.5 \quad \forall\, i \geq 1. \tag{4}$$

In the setting of this paper, it is important to distinguish between subjects (such as users) and events generated by subjects (such as orders). Throughout the paper, we work with variables associated with events and not subjects. For example, $Y_i$, $i \geq 1$, are event outcomes and $W_i$, $i \geq 1$, are event assignment indicators. One implication of this is that even though the subjects are assigned to one of the two groups independently at random, the variables associated with individual events can be dependent (as some of them can be generated by the same user). Therefore, in our theory (Theorems 1 and 2), we allow the event outcomes $Y_i$, $i \geq 1$, to be dependent and, likewise for the event assignment indicators $W_i$, $i \geq 1$. Part (a) of the null hypothesis stated above only assumes that the two sets of variables ($\{Y_1, Y_2, \ldots\}$ and $\{W_1, W_2, \ldots\}$) are independent from each other, but the variables within each of the two sets are allowed to be dependent.

The null hypothesis stated above exhibits the difference between our setup and the classical testing for a zero average treatment effect. Specifically,

- our testing procedure acts on the level of events generated by experiment subjects and not the experiment subjects themselves (for example, orders and not customers);

- we assume that under the null hypothesis, the treatment does not affect the "event generation process" in any way.

In the revenue monitoring example, the latter implies that both the frequency of orders and the order value distribution stay the same in both groups under the null. We discuss the limitations of the considered setup in Section 5.

The approach described so far allows us to derive a simple but effective sequential testing procedure we present next.

## 3.3 Proposed Sequential Testing Method

The proposed monitoring method consists of tracking the difference in the metric of interest between the experiment groups and (continuously) comparing it with a constant (alerting) boundary. Whenever the boundary is crossed, we can conclude (with a predefined significance level $\alpha$) that the treatment has an impact on the metric.

The procedure requires setting the maximum number of events to be observed during the monitoring period that we denote by $N$. In other words, the boundary is computed assuming that the monitoring will not continue after $N$ events have been collected. See Section 3.3.2 for how to set $N$ in practice.

The proposed method has only one other parameter that needs to be estimated beforehand - the (scaled) variance of the difference in the metric of interest at the end of monitoring, $V_N = var(S_N)/N$. The estimator of $V_N$ needs to be consistent under the null hypothesis. Given the estimate $\hat{V}_N$ (computed by such an estimator), the alerting boundary is set to

$$b^* = z_{1-\alpha/2} \cdot \sqrt{N\hat{V}_N}, \tag{5}$$

where $z_{1-\alpha/2}$ is the quantile of the standard normal distribution of level $1 - \alpha/2$. The proposed simple sequential test proceeds by tracking $S_n$ and comparing it against $b^*$.

The necessity to fix or estimate $N$ and to provide an estimate of $V_N$ beforehand can be perceived as a disadvantage of the proposed approach. However, we argue that in many practical scenarios, those parameters can be reliably set based on pre-experiment data (in fact, the underlying data distribution has to have some level of stability or regularity in time for the A/B testing to be meaningful in the first place).

### 3.3.1 Informal Derivation of the Proposed Method

The derivation of the alerting boundary (5) follows two steps. First, we use (generalised) Levy's inequality[1] to bound the false detection rate by the probability that the tracked sum exceeds the threshold at the end of monitoring, i.e.,

$$FDR = \Pr\left\{\max_{n=1}^{N} S_n > b\right\} \leq 2\Pr\left\{S_N > b\right\}. \tag{6}$$

In the second step we use the Central Limit Theorem to approximate the latter probability,

$$\Pr\left\{S_N > b\right\} \approx 1 - \Phi\left(\frac{b}{\sqrt{var(S_N)}}\right) = 1 - \Phi\left(\frac{b}{\sqrt{NV_N}}\right), \tag{7}$$

where $\Phi$ denotes the CDF of the standard normal distribution. Then by setting the threshold $b$ to $z_{1-\alpha/2} \cdot \sqrt{NV_N}$ (cf. (5)), we ensure that the false detection rate is under control, i.e., $FDR \lessgtr \alpha$. This is formalised in Theorem 1 below.

The proposed testing procedure is defined in Algorithm 1 and will be referred to as *YEAST* (from YEt Another Sequential Test). The estimation of $V_N$ is discussed in Section 3.3.2 below.

---
**Algorithm 1** YEAST
---
**Require:** $N, \hat{V}_N$
  $b^* \leftarrow z_{1-\alpha/2}\sqrt{N\hat{V}_N}$
  **for** $n = 1, \ldots, N$ **do**
    **if** $S_n > b^*$ **then**
      flag significance
    **end if**
  **end for**

---

For a two-sided test, we track $|S_n|$ and compare it against $b^*_{2-sided} = z_{1-\alpha/4} \cdot \sqrt{N\hat{V}_N}$.

Mind that differently from the non-sequential testing, the significance level $\alpha$ needs to be divided by 2 for the one-sided test and by 4 for the two-sided test. This is due to the factor of two in the right-hand side of bound (6).

Note that while baring some resemblance in the formulas, YEAST is fundamentally different from GST [8] methods. The latter use a statistic normalized by the observed information (i.e., the variance of the cumulative data up to time $n$) and adjust the rejection threshold at each look. In contrast, our method normalizes the partial cumulative sum $S_n$ by the variance of the full dataset $var(S_N)$ and compares it to a fixed threshold. As a result, the boundary of YEAST is constant (does not change from one look to another) while this is not the case for GST. The computation of the YEAST boundary is computationally simple and does not require numerical integration.

---
[1]We leverage a generalised version of the inequality from [21, Paragraph 29.1, A] that allows the observations to be dependent.

### 3.3.2 Setting the Input Parameters

As was described in the previous subsection, the computation of the alerting boundary of the proposed sequential testing procedure requires two inputs: the number of events $N$ to be observed during the monitoring and an estimate $\hat{V}_N$ of the (scaled) variance of the tracked metric difference at the end of the monitoring. In this subsection, we discuss how to provide those inputs in practice.

To set $N$, we estimate the expected number of events to be collected over the experiment time frame. In the simplest form, this can amount to setting $N$ to the number of events observed in a pre-experiment period of the same duration as the experiment we want to monitor.

For variance estimation, we suggest using pre-experiment data as well. Obtaining an accurate estimate this way requires the pre-experiment period to be representative of the experiment time. However, in cases where this cannot be assumed, the validity of fixed-duration randomised experiments is questionable in principle (as the effects measured during such experiments may not generalise beyond their time frames). So in the context of A/B testing, this is a natural assumption. Note that estimating variance from pre-experiment data is a standard procedure performed before an A/B test because it is used in computing the MDE (minimum detectable effect) and identifying the necessary sample size.

As noted in Section 3.2, the event outcomes $Y_i$, $i \geq 1$, can be dependent because some of the events are generated by the same user. The same holds for the event assignment indicators $W_i$, $i \geq 1$. Consequently, the increments $X_i$, $i \geq 1$, of the monitored trajectory can be serially correlated, in which case the variance $var(S_N)$ does not equal the sum of the variances $var(X_i)$. The serial correlation of the increments has a clustered nature in this case because variables corresponding to different users are independent due to random assignment. Therefore, we propose to use an estimator that is robust to clustered serial correlation [1] to estimate $V_N$. Implementations of such estimators are readily available, for example as part of the `sandwich` package in R [41, 40] (see function `vcovCL`). The cluster variable in our case is the user identifier. As we show in our experiments with real-world data in Section 4, the use of a robust (clustered) variance estimator can be utterly important for effective control of the false detection rate.

Although the need to set a finite horizon $N$ can be viewed as a disadvantage of YEAST, we have not found it problematic in practice. When running A/B experiments it is typical to plan for a certain duration, and the number of observations collected during the experiment time frame is usually predictable. In addition, the decision boundary of YEAST depends on the square root of $N$, which makes the method more robust to inaccuracies in the estimation of $N$. That said, we acknowledge that there can be situations where it is difficult to reliably estimate $N$ and, in that case, it may be better to resort to other methods (e.g., [11]) that do not require setting a finite horizon. In the empirical evaluations in Section 4, we never assume that $N$ is known and estimate it from pre-experiment data (as part of the method). Hence, the respective empirical results incorporate the uncertainty in the estimation of $N$.

### 3.3.3 Formal Statements

In this subsection, we will justify the correctness of the proposed testing procedure. This includes a statement demonstrating that the proposed method controls the false detection rate (Theorem 1), and a statement showing that the power of the method is asymptotically one (Theorem 2).

In the formal statements below, $(\Omega, \mathcal{F}, \mathrm{Pr})$ is a probability space and $\mathbb{R}$ stands for the set of real numbers. As before, $\Phi$ denotes the CDF of the standard normal distribution. The proofs of the statements can be found in Appendix A.

The first theorem states that when the treatment has no actual impact, detections occur sufficiently rarely.

**Theorem 1** (False Detection Rate Control). *Let $Y_i \colon \Omega \mapsto \mathbb{R}$ and $W_i \colon \Omega \mapsto \{0, 1\}$, $i = 1, \ldots, N$, be random variables and let $S_n$, $n \geq 1$, be the running sum defined by (1)–(2). Suppose that Assumption 1 and the null hypothesis (3)–(4) hold and that a version of the Central Limit Theorem is valid for the sequence of random variables $X_1, X_2, \ldots$ defined by (1). Assume that $var(S_N) > 0$ for all $N \geq 1$ and set $V_N := var(S_N)/N$.*

*Then for any $\gamma > 0$ and any $\varepsilon > 0$ there exists $N_{\gamma, \varepsilon}$ such that for all $N \geq N_{\gamma, \varepsilon}$ we have*

$$\mathrm{Pr}\left\{ \max_{n=1}^{N} S_n > \gamma \sqrt{N V_N} \right\} \leq 2\left(1 - \Phi\left(\gamma\right)\right) + \varepsilon.$$

*In particular, for any significance level $\alpha \in (0, 1)$ and any $\varepsilon > 0$ the false detection rate* $\Pr\left\{\max_{n=1}^{N} S_n > z_{1-\alpha/2}\sqrt{NV_N}\right\}$ *is bounded by $\alpha + \varepsilon$, provided that $N$ is sufficiently large.*

Note that we deliberately do not spell out specific conditions, ensuring that the central limit properties hold for the sequence of increments $\{X_i\}$. Multiple sets of such conditions exist. A good overview of them can be found in [38, Chapter V].

The above theorem not only justifies the proposed method, but also provides insight into how the quality of the false detection rate control depends on the error in setting the input parameters of the method ($N$ and/or $V_N$). Specifically, if we overestimate or underestimate the product $NV_N$ by a factor of $r$ the effective threshold used during monitoring will equal $z_{1-\alpha/2}\sqrt{r}\sqrt{NV_N}$ and, as follows from (1), the false detection rate will be (approximately) bound by $2\left(1 - \Phi\left(z_{1-\alpha/2}\sqrt{r}\right)\right)$ instead of $\alpha$. For example, if $NV_N$ is overestimated by 20% the bound becomes equal approximately 3% instead of the nominal significance level of 5%.

Our second theorem claims that, when treatment has an effect, its detection can be made as certain as desired if we monitor for sufficiently long.

**Theorem 2** (Power). *Let $Y_i\colon \Omega \mapsto \mathbb{R}$ and $W_i\colon \Omega \mapsto \{0, 1\}$, $i = 1, \ldots, N$, be random variables, and let $S_n$, $n \geq 1$, be the running sum defined by (1)–(2). Suppose that Assumption 1 holds and a version of the Central Limit Theorem is valid for the sequence of random variables $X_1, X_2, \ldots$ defined by (1). Assume that $var(S_N) > 0$ for all $N \geq 1$ and that the set of variables $V_N = var(S_N)/N$, $N \geq 1$, is stochastically bounded. Finally, let $E[X_i] = \mu > 0$ (implying that the null hypothesis (3)–(4) is violated). Then $\Pr\left\{\max_{n=1}^{N} S_n > z_{1-\alpha/2}\sqrt{NV_N}\right\} \to 1$ as $N \to \infty$*

## 4 Validation Using Real-World Data: a Semi-Synthetic Experiment

The best possible real-world validation would be using data from a very large number of A/B tests (some with statistically significant results and some with neutral outcomes under high power). Having those data, one could apply the compared sequential tests retrospectively and measure their false detection rates and sensitivity (power). As we do not have access to a sufficient number of past A/B tests, we chose a different approach and conducted a semi-synthetic study, in which we took a public real-world data set ("Online Retail" dataset [5]), randomly assigned users in that data set to control and test, computed the cumulative difference in the metric of interest (revenue) between control and treatment for each assignment replication, and measured the associated detection rates. (For power assessment, we also artificially decreased the order values considering multiple effect sizes, see Section 4.2 below.)

The code implementing our experiments is openly available in the git repository[16].

Here are the methods we compared.

**YEAST** The proposed sequential method using the significance boundary (5) as presented in Section 3.

**mSPRT** The mixture sequential probability ratio test [27, 20]. We set the tuning parameter of the method to 11, 25, and 100 and denote the corresponding versions as mSRTphi11, mSRTphi25, and mSRTphi50.

**GAVI** The generalization of the always valid inference, as proposed in [11]. As in [28], we set the numerator of parameter $\rho$ of the method to 250, 500, and 750 and denote the corresponding instances of the method by GAVI250, GAVI500, and GAVI750, respectively. We also included GAVI with the tuning parameter $\rho$ set to 10,000 (the default setting used by Eppo).

**LanDeMetsOBF** The GST method [8] with the O'Brien-Fleming alpha-spending function [23]. For computational reasons, we constructed the boundary using 100 interim checkpoints and then extended it in a piecewise-constant manner to support continuous monitoring.

**SeqC2ST-QDA** The sequential predictive test from [25] with the online Newton step (ONS) strategy for selecting betting fractions. We used QDA (quadratic discriminant analysis) as a classification model for this method.

### 4.1 Type-I Error (False Detection Rate)

The public dataset [5] consists of all transactions that occurred between 2010-12-01 and 2011-12-09 on a UK-based online store. We used the data up until 2011-11-30 to have 12 complete months and dropped orders with missing customer identifiers. The resulting dataset had 21,269 orders. We split it into two halves: the first 6 months were used for estimating parameters ($N$ and $V_N$) and the latter 6 months - for validation. Using the validation period, we randomly assigned customers to control and treatment 100,000 times[2], computed the cumulative difference in the revenue between control and treatment for each assignment replication, applied the sequential tests to monitor the cumulative difference, and measured the frequency of detections over the replications. Since no actual treatment was applied to the replicated treatment groups, the null hypothesis was true, and the computed rates were false detection rates.

It is typical of online transaction data to contain outliers (e.g., large wholesale customers that can easily skew the monitored metric difference in favour of one of the groups). Therefore, before replicating the assignments, we removed outliers via revenue capping at the 99.9th percentile of the total revenue generated by a customer. The capping was applied progressively as follows: a given customer was only monitored while his/her revenue stayed within the cap and all subsequent events *from that customer* were discarded.

The measured detection rates are reported in Table 2 together with the associated confidence intervals (computed with Bonferroni correction). Detection rates within one percentage point of the nominal significance level (set to 5%) are shown in bold.

As mentioned in Section 3.1, we employed sequential tests with two different variance estimates. One of the two assumed that the increments $X_i$ of the monitored difference-in-sum are independent. The other was an estimator robust to cluster serial correlation, that is — to the correlation between increments coming from the same user in our case.[3] The results corresponding to the use of the two estimators are reported in columns 'non-robust' and 'robust' of Table 2, respectively.[4]

Two observations from our experiments are in order. Firstly, the proposed method (YEAST) with the robust variance estimator was the only method to demonstrate accurate false detection rate control (i.e., with the detection rate staying within one percentage point of the nominal significance level). Secondly, when the non-robust estimator was used, the detection rate was considerably inflated. This highlights the importance of accounting for the correlation in the events generated by the same user when estimating the variance.

Table 2: False Detection Rate

| | | | |
|---|---|---|---|
| 1 | YEAST | **0.044 ± 0.002** | 0.167 ± 0.004 |
| 2 | mSPRT100 | 0.028 ± 0.002 | 0.159 ± 0.004 |
| 3 | mSPRT11 | 0.017 ± 0.001 | 0.127 ± 0.003 |
| 4 | mSPRT25 | 0.022 ± 0.001 | 0.141 ± 0.003 |
| 5 | GAVI250 | 0.024 ± 0.002 | 0.148 ± 0.003 |
| 6 | GAVI500 | 0.027 ± 0.002 | 0.156 ± 0.004 |
| 7 | GAVI750 | 0.029 ± 0.002 | 0.160 ± 0.004 |
| 8 | GAVI10K | 0.028 ± 0.002 | 0.160 ± 0.004 |
| 9 | LanDeMetsOBF | 0.026 ± 0.006 | 0.133 ± 0.011 |
| 10 | SeqC2ST-QDA | 0.102 ± 0.003 | |

---

[2]We generated the assignments in parallel using 13 processes. The random seed of the $i$-th process was set to $i$. Each process but the last made 7692 replications while the last one made 7696 so the total number of replications was 100,000. The simulations take about 15 min to run on a laptop with an Apple M3 Pro CPU and 18 GB RAM.

[3]We used the `vcovCL` function of the `sandwich` R package to compute the robust variance estimate. To reduce the sparsity in the time dimension, we summed the data (revenue increments) by user-hour before computing the estimator. See the linked repository [16] for details.

[4]SeqC2ST-QDA does not take an estimate of the increment variance as input hence there is only one result for this method.

## 4.2 Type-II Error (Power) Assessment

To assess power, we conducted the same simulations as in Section 4.1 but additionally decreased the values of order values belonging to the treatment group in each replication. The relative decrease was equal to 5, 10, and 20%.

To evaluate the sensitivity of the tests, we included an additional baseline: the standard non-sequential Student's t-test, applied once at the end of each experiment replication using the full dataset. This test serves as a benchmark because it is widely used in practice and represents the level of power achievable when no early stopping is employed (and thus no adjustments are needed to control the false detection rate).

The reported results are for the robust variance estimate. The detection rates can be found in Table 3. As can be seen, the proposed method (YEAST) is the only method that was not underpowered relative to the standard (non-sequential) Student's t-test.

Table 3: Semi-synthetic Experiment: Power (Online Retail)

| | method | relative decrease in the order value | | | |
|---|---|---|---|---|---|
| | | 0.05 | 0.1 | 0.2 | 0.5 |
| | *Non-seq. ttest* | *0.115 ± 0.004* | *0.253 ± 0.005* | *0.686 ± 0.005* | *1.000 ± 0.000* |
| 1 | **YEAST** | **0.116 ± 0.004** | **0.250 ± 0.005** | **0.678 ± 0.005** | **1.000 ± 0.000** |
| 2 | mSPRT100 | 0.006 ± 0.001 | 0.018 ± 0.002 | 0.139 ± 0.004 | 0.998 ± 0.001 |
| 3 | mSPRT011 | 0.002 ± 0.001 | 0.007 ± 0.001 | 0.072 ± 0.003 | 0.993 ± 0.001 |
| 4 | mSPRT025 | 0.003 ± 0.001 | 0.010 ± 0.001 | 0.093 ± 0.003 | 0.996 ± 0.001 |
| 5 | GAVI250 | 0.003 ± 0.001 | 0.012 ± 0.001 | 0.107 ± 0.003 | 0.997 ± 0.001 |
| 6 | GAVI500 | 0.005 ± 0.001 | 0.017 ± 0.001 | 0.131 ± 0.004 | 0.998 ± 0.001 |
| 7 | GAVI750 | 0.006 ± 0.001 | 0.019 ± 0.002 | 0.146 ± 0.004 | 0.999 ± 0.000 |
| 8 | GAVI10K | 0.012 ± 0.001 | 0.038 ± 0.002 | 0.232 ± 0.005 | 1.000 ± 0.000 |
| 9 | LanDeMetsOBF | 0.076 ± 0.003 | 0.180 ± 0.004 | 0.584 ± 0.005 | 1.000 ± 0.000 |
| 10 | SeqC2ST-QDA | 0.091 ± 0.003 | 0.090 ± 0.003 | 0.094 ± 0.003 | 0.107 ± 0.003 |

## 5 Limitations

We saw in Section 4 that despite its simplicity, the proposed monitoring procedure demonstrated effectiveness in controlling type-I error and a higher power relative to existing SOTA approaches. Yet there are scenarios where the underlying null hypothesis of no effect on the data generation process is too strong, potentially leading to some "undesirable sensitivity". In other words, sometimes the treatment can alter the data generation process but have a neutral average effect on the metric of interest. We can think of (at least) two such scenarios.

*Situations where the treatment brings multiple changes to the data generation process and those changes counter-balance each other leading to a neutral average effect on the metric.* For example, the treatment can make users place fewer orders but increase the average order value at the same time so the two effects neutralise each other and the revenue per user stays intact.

*Situations where the treatment changes the variance of the event outcome, leaving the average unaffected.* For example, the treatment can increase the variability in the order values without changing their average.

If in a given practical application, having sensitivity to one of the scenarios above is highly undesirable, it is safer to adhere to alternative sequential testing methods (e.g., [11, 10]).

Note that none of the problematic two scenarios can materialise when the event outcome is constant (i.e., the metric of interest simply counts events). In addition, the first scenario cannot hold when the treatment cannot affect the number of events, for example, when the events correspond to incoming patients in a medical study.

From a theoretical perspective, the amount of oversensitivity of the proposed method in the above two situations would depend on how much the median of the "residual sum" (or the sum of the remaining increments until the end of monitoring) deviates from zero. Intuitively, in the early steps of monitoring, this should not be a problem because the number of remaining increments is large and their sum's distribution would be approximately symmetric due to the Central Limit Theorem.

Yet, closer to the end of monitoring, this deviation can be considerable. Mitigation strategies can include stopping the monitoring early or setting the threshold as if the monitoring is to be run for longer than in reality. The effectiveness of those mitigation strategies and the tradeoff with the power would depend on how quickly the distribution of increment sums converges to a normal distribution.

We see studying the behaviour of our proposed method in situations mentioned in this section and seeking mitigation strategies as a large topic for future research.

It is worth emphasising that in some situations, it can be desirable to capture all relevant potential defects (beyond the effect on the mean) [20]. For example, an increased outcome variance can indicate that the effect of the treatment is heterogeneous and some important subgroups of the population can be negatively affected (even though the overall mean stays the same). In such scenarios, the extra sensitivity mentioned in this section is not really a limitation but an advantage instead.

## 6   Conclusion

This paper proposes a novel sequential test for continuous experiment monitoring. The proposed test supports both discrete and real-valued metrics. Its effectiveness is demonstrated in an empirical study (a semi-synthetic experiment based on real-world data). We hope that the proposed method will enable a wider use of sequential testing for online evaluation.

## Acknowledgments and Disclosure of Funding

The authors are grateful to Evan Miller for insightful comments and his excellent blog [22], which sparked the idea for this research.

This research was carried out while all the authors worked at Zalando. The paper does not represent, imply, or establish any form of partnership, joint venture, or collaboration — whether formal or informal — between the organizations with which the authors are affiliated at present.

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

# Appendix

## A   Proofs and Auxiliary Statements

**Lemma 1.** *Let $Y_i \colon \Omega \mapsto \mathbb{R}$ and $W_i \colon \Omega \mapsto \{0, 1\}$, $i = 1, \ldots, N$, be random variables, and let $S_n$, $n \geq 1$, be the running sum defined by* (1)–(2). *Suppose that Assumption 1 and the null hypothesis* (3)–(4) *hold.*

*Then for any $N \geq 1$ and any $k = 1, \ldots, N$ the distribution of $S_N - S_k$ conditional on $S_1, \ldots, S_k$ is symmetric around 0.*

*Proof.* Fix any $N \geq 2$ and any $k = 1, \ldots, N-1$ (for $N = 1$ or $k = N$ the statement is trivial). Let $\mathcal{T} = \{-1, +1\}^{N-k}$ be the set of all possible combinations of $-1$ and $+1$ of length $N - k$. Furthermore, for any $\tau \in \mathcal{T}$, let $\bar{\tau} \in \mathcal{T}$ be the combination obtained by negating every element of $\tau$, i.e., $\bar{\tau}_j = -\tau_j$ for all $j = 1, \ldots, N - k$. Next, define

$$A_\tau = \{1 - 2W_i = \tau_{i-k} \,\forall\, i = k+1, \ldots, N\}, \quad \tau \in \mathcal{T}.$$

Note that

$$A_\tau \cap A_{\tau'} = \emptyset \,\forall\, \tau, \tau' \in \mathcal{T} \text{ and } \cup_{\tau \in \mathcal{T}} A_\tau = \cup_{\tau \in \mathcal{T}} A_{\bar{\tau}} = \Omega. \tag{8}$$

Also, Assumption 1 together with the null hypothesis imply that

$$\Pr(A_\tau \mid Y_1, W_1, \ldots, Y_k, W_k) = \Pr(A_{\bar{\tau}} \mid Y_1, W_1, \ldots, Y_k, W_k) \quad \text{a.s.} \quad \forall\, \tau \in \mathcal{T}. \tag{9}$$

Finally, fix an arbitrary Borel set $Q \subset \mathbb{R}$ and define

$$B = \{S_N - S_k \in Q\}, \quad \bar{B} = \{S_k - S_N \in Q\}.$$

Denoting the conditional probability $\Pr(\cdot \mid Y_1, W_1, \ldots, Y_k, W_k)$ by $\Pr_k$, we will now derive the key relationship for the proof of the lemma. Specifically, (3) and (9) imply that for any $\tau \in \mathcal{T}$ we have[5]

$$
\begin{aligned}
\Pr_k(\bar{B} \cap A_\tau) &= \Pr_k\left(\left\{-\sum_{i=k+1}^{N} \tau_{i-k} Y_i \in Q\right\} \cap A_\tau\right) \\
&\overset{(3)}{=} \Pr_k\left\{-\sum_{i=k+1}^{N} \tau_{i-k} Y_i \in Q\right\} \Pr_k(A_\tau) \\
&\overset{(9)}{=} \Pr_k\left\{-\sum_{i=k+1}^{N} \tau_{i-k} Y_i \in Q\right\} \Pr_k(A_{\bar{\tau}}) \\
&= \Pr_k\left\{\sum_{i=k+1}^{N} \bar{\tau}_{i-k} Y_i \in Q\right\} \Pr_k(A_{\bar{\tau}}) \\
&\overset{(3)}{=} \Pr_k\left(\left\{\sum_{i=k+1}^{N} \bar{\tau}_{i-k} Y_i \in Q\right\} \cap A_{\bar{\tau}}\right) \\
&= \Pr_k(B \cap A_{\bar{\tau}}) \quad \text{a.s.}
\end{aligned}
\tag{10}
$$

From (10) and (8), it follows that

$$\Pr_k(\bar{B}) \overset{(8)}{=} \sum_{\tau \in \mathcal{T}} \Pr_k(\bar{B} \cap A_\tau) \overset{(10)}{=} \sum_{\tau \in \mathcal{T}} \Pr_k(B \cap A_{\bar{\tau}}) \overset{(8)}{=} \Pr_k(B) \quad \text{a.s.,}$$

which by the tower property implies that

$$\Pr(\bar{B} \mid S_1, \ldots, S_k) = \Pr(B \mid S_1, \ldots, S_k) \quad \text{a.s.} \tag{11}$$

Since the fixed Borel set $Q$ was arbitrary, (11) proves that conditionally on $S_1, \ldots, S_k$, the distribution of $S_N - S_k$ and $S_k - S_N$ is the same or, equivalently, the distribution of $S_N - S_k$ is symmetric about zero. $\square$

---

[5]In the second and next to the last transitions, we leverage the fact that if two collections of random variables are independent, they remain conditionally independent after conditioning on any subset of variables from either or both collections.

**Lemma 2.** *Suppose that the assumptions of Lemma 1 hold.*

*Then for any $N \geq 1$ and any $b > 0$, we have*

$$\Pr\left\{\max_{n=1}^{N} S_n \geq b\right\} \leq 2\Pr\{S_N \geq b\}, \tag{12}$$

*and*

$$\Pr\left\{\max_{n=1}^{N} |S_n| \geq b\right\} \leq 2\Pr\{|S_N| \geq b\}. \tag{13}$$

*Proof.* From Lemma 1, it follows that for any $k = 1, \ldots, N$, the distribution of $S_N - S_k$ conditional on $S_1, \ldots, S_k$ is symmetric about 0 and therefore the conditional median $\mu(S_N - S_k \mid S_1, \ldots, S_k)$ is zero. Then (12) and (13) easily follow from the generalised Levy's inequalities [21, Paragraph 29.1, A]. □

*Proof of Theorem 1.* This proof has two steps. First, we bound the chance of "crossing the threshold" at any moment during the monitoring by twice the probability of crossing it at the last moment. Secondly, we approximate the latter probability with the help of the Central Limit Theorem (noting that the tracked difference at the end of monitoring is the sum of a large number of increments). This allows us to construct the threshold appropriately, ensuring that the method's false detection rate is small enough.

Fix an arbitrary $\gamma > 0$ and $\varepsilon > 0$.

By the continuity of $\Phi$, there exists a sufficiently small $\delta > 0$ such that

$$|\Phi(\gamma) - \Phi(\gamma - \delta)| \leq \frac{\varepsilon}{4}. \tag{14}$$

Furthermore, the null hypothesis implies that

$$\mathrm{E}[X_i] = \mathrm{E}\left[(1 - 2W_i)Y_i\right] \stackrel{(3)}{=} (1 - 2\Pr\{W_i = 1\})\,\mathrm{E}[Y_i] \stackrel{(4)}{=} 0$$

for all $i \geq 1$. Therefore, $E[S_N] = 0$ for all $N \geq 1$ and by the Central Limit Theorem we have that

$$\left|\Pr\left\{S_N/\sqrt{var(S_N)} \leq \gamma - \delta\right\} - \Phi(\gamma - \delta)\right| \leq \frac{\varepsilon}{4} \tag{15}$$

for all sufficiently large $N$. (12), relations (14)–(15) give

$$
\begin{aligned}
\Pr\left\{\max_{n=1}^{N} S_n > \gamma\sqrt{NV_N}\right\} \quad &= \quad \Pr\left\{\max_{n=1}^{N} S_n > \gamma\sqrt{var(S_N)}\right\} \\
&\leq \quad \Pr\left\{\max_{n=1}^{N} S_n \geq \gamma\sqrt{var(S_N)}\right\} \\
&\stackrel{(12)}{\leq} \quad 2\Pr\left\{S_N \geq \gamma\sqrt{var(S_N)}\right\} \\
&= \quad 2\Pr\left\{S_N/\sqrt{var(S_N)} \geq \gamma\right\} \\
&\leq \quad 2\Pr\left\{S_N/\sqrt{var(S_N)} > \gamma - \delta\right\} \\
&= \quad 2\left(1 - \Pr\left\{S_N/\sqrt{var(S_N)} \leq \gamma - \delta\right\}\right) \\
&= \quad 2\left(1 - \Phi(\gamma)\right) + 2\left(\Phi(\gamma) - \Phi(\gamma - \delta)\right) \\
&\qquad + 2\left(\Phi(\gamma - \delta) - \Pr\left\{S_N/\sqrt{var(S_N)} \leq \gamma - \delta\right\}\right) \\
&\stackrel{(14),\,(15)}{\leq} \quad 2\left(1 - \Phi(\gamma)\right) + \frac{\varepsilon}{2} + \frac{\varepsilon}{2} \\
&= \quad 2\left(1 - \Phi(\gamma)\right) + \varepsilon
\end{aligned}
$$

for all large enough $N$. In particular, for any $\alpha \in (0, 1)$ and any $\varepsilon > 0$ we have that

$$\Pr\left\{\max_{n=1}^{N} S_n > z_{1-\alpha/2}\sqrt{NV_N}\right\} \leq 2\left(1 - \Phi\left(z_{1-\alpha/2}\right)\right) + \varepsilon = \alpha + \varepsilon$$

for all sufficiently large $N$. This completes the proof. □

*Proof of Theorem 2.* The idea of the proof is to note that the probability of detecting the effect at any moment during the monitoring is at least as high as the probability of detecting it at the very end. Then we leverage the Central Limit Theorem to show that (when the treatment effect exists) the latter probability approaches 1 as the monitoring duration increases.

Let $\xi_N$ stand for the centered and standardised sum $(S_N - N\mu)/var(S_N)$. By the Central Limit Theorem $\xi_N \xrightarrow{d} \mathcal{N}(0, 1)$ and therefore $\xi_N$ is stochastically bounded (see, for example, [34, Theorem 2.4]).

Now, fix an arbitrary $\varepsilon > 0$. From the stochastical boundedness of $\xi_N$ and $V_N$, it follows that there exist $M_1$ and $M_2$ such that for all sufficiently large $N$ we have

$$\Pr\{|\xi_N| \leq M_1\} \geq 1 - \frac{\varepsilon}{2} \tag{16}$$

and

$$\Pr\{|V_N| \leq M_2\} \geq 1 - \frac{\varepsilon}{2}. \tag{17}$$

From (16)–(17), it follows that

$$\Pr\left\{\xi_N + \mu\sqrt{N}/\sqrt{V_N} \geq -M_1 + \mu\sqrt{N}/\sqrt{M_2}\right\} \geq 1 - \varepsilon$$

for all large enough $N$. But then for any sufficiently large $N$, we have that

$$\Pr\left\{\xi_N + \mu\sqrt{N}/\sqrt{V_N} > z_{1-\alpha/2}\right\} \geq 1 - \varepsilon. \tag{18}$$

because $\left(-M_1 + \mu\sqrt{N}/\sqrt{M_2}\right) \to \infty$ as $N \to \infty$. Then we can derive that

$$
\begin{aligned}
\Pr\left\{\max_{n=1}^{N} S_n > z_{1-\alpha/2}\sqrt{NV_N}\right\} &\geq \Pr\left\{S_N > z_{1-\alpha/2}\sqrt{NV_N}\right\} \\
&= \Pr\left\{\xi_N + \mu\sqrt{N}/\sqrt{V_N} > z_{1-\alpha/2}\right\} \\
&\overset{(18)}{\geq} 1 - \varepsilon
\end{aligned}
$$

for all sufficiently large $N$. Since $\varepsilon$ was arbitrary, this completes the proof. $\square$

# B   Experiments using synthetic data

Our simulation setup follows a recent study [28]. To the best of our knowledge, this is the only existing comparative study of sequential tests. We extended its implementation [29] with the method proposed in Section 3 of the paper (YEAST).

As in [29], we generate $N$ observations from the control, $Y_i^c$, $i = 1, \ldots, N$, and $N$ observations from treatment, $Y_i^t$, with $N$ set to 500. Both $Y_i^c$ and $Y_i^t$ are generated as IID random variables. The effect size (on the mean) took values 0.0, 0.1, 0.2, 0.3, and 0.4. The $X_i$ variables (increments in the metric difference between the two groups) were computed as $X_i = Y_i^c - Y_i^t$, $i = 1, \ldots, N$.

We conducted two simulation experiments. The first experiment repeats the simulation study from [28] but adds YEAST to the comparison. In the second set of simulations, we explored the behaviour of the compared methods when the increments are non-normal.

In all experiments, the target significance level was set at 5% and the number of replications was set to 100,000.

The simulation code for these experiments can be found in the associated git repository [16].

## B.1   Simulation Results

In this experiment, observations $Y_i^c$ and $Y_i^t$ were drawn from normal distributions with parameters $(1, 1)$ and $(1 + \xi, 1)$, respectively. The effect size $\xi$ took values 0.0, 0.1, 0.2, 0.3, and 0.4. The random generator seed was set to 8163 (as in [28, 29]).

In the following we present the list of methods we compared.

**YEAST** The proposed sequential method.

**YEASTnv{K}** (with $K = 80, 90, 110, 120$) are the instances of the proposed method with the product $NV_N$ misestimated by a factor of 0.8, 0.9, 1.1 and 1.2, respectively (ie the method is applied with $NV_N$ set at 10% and 20% below and above the true value). Since the alerting boundary of YEAST depends on the product of $N$ and $V_N$ we can study the effect of inaccuracies in their estimation together.

**mSPRT** The mixture sequential probability ratio test [27, 20]. We set the tuning parameter of the method to 11, 25, and 100 and denote the corresponding versions as mSRTphi11, mSRTphi25, and mSRTphi50.

**GAVI** The generalization of the always valid inference, as proposed in [11]. As in [28], we set the numerator of parameter $\rho$ of the method to 250, 500, and 750 and denote the corresponding instances of the method by GAVI250, GAVI500, and GAVI750, respectively.

**LanDeMetsOBF** The GST method [8] with the O'Brien-Fleming alpha-spending function [23]. For computational reasons, we constructed the boundary using 100 interim checkpoints and then extended it in a piecewise-constant manner to support continuous monitoring.

**SeqC2ST-QDA** The sequential predictive test from [25] with the online Newton step (ONS) strategy for selecting betting fractions. We used QDA (quadratic discriminant analysis) as a classification model for this method.

**Bonferroni** A naive approach using Bonferroni corrections.

All of the compared methods were employed in the continuous monitoring mode meaning that the check for significance was performed after each observation. In Appendix C we report additional evaluation results for the case where YEAST was employed in the "discrete mode" (i.e., with a fixed number of interim significance checks).

For each experimental setting, we conducted 100,000 replications. Each replication can result in a detection or no-detection. A detection occurs when the respective test flags significance (at any point of the monitoring process). We compute the share of replications where a detection occurs. When the treatment does not have an effect this share is the so-called (empirical) false detection rate (or, synonymously, false positive rate, type-I error, or test "size"). In settings where the treatment has an effect, this share is the (empirical) power. Table 4 presents the measured false detection rate and power, along with their corresponding 95% confidence intervals adjusted for multiple comparisons.

The methods that keep the false detection rate below the nominal level of 5% and have the highest power are shown in bold.

One can see from the table that YEAST demonstrated the highest power among the continuous monitoring approaches that did not inflate the false detection rate. It means that while it kept the false positive rate under control when there was no treatment effect, YEAST had the highest sensitivity among the compared approaches when the treatment had an effect on the metric of interest.

Table 4: Simulation Experiment:False Detection Rate and Power

| | effect size | 0.0 | 0.1 | 0.2 | 0.3 |
|---|---|---|---|---|---|
| | method | | | | |
| 1 | YEAST | $0.047 \pm 0.002$ | $\mathbf{0.448 \pm 0.005}$ | $\mathbf{0.902 \pm 0.003}$ | $\mathbf{0.995 \pm 0.001}$ |
| 2 | YEASTn110 | $0.038 \pm 0.002$ | $0.408 \pm 0.005$ | $0.883 \pm 0.003$ | $\mathbf{0.994 \pm 0.001}$ |
| 3 | YEASTn120 | $0.030 \pm 0.002$ | $0.372 \pm 0.004$ | $0.864 \pm 0.003$ | $0.992 \pm 0.001$ |
| 4 | YEASTn80 | $0.075 \pm 0.002$ | $0.534 \pm 0.005$ | $0.933 \pm 0.002$ | $0.997 \pm 0.000$ |
| 5 | YEASTn90 | $0.059 \pm 0.002$ | $0.490 \pm 0.005$ | $0.918 \pm 0.003$ | $0.996 \pm 0.001$ |
| 6 | mSPRT100 | $0.016 \pm 0.001$ | $0.235 \pm 0.004$ | $0.751 \pm 0.004$ | $0.976 \pm 0.001$ |
| 7 | mSPRT011 | $0.032 \pm 0.002$ | $0.249 \pm 0.004$ | $0.739 \pm 0.004$ | $0.972 \pm 0.002$ |
| 8 | mSPRT025 | $0.028 \pm 0.002$ | $0.260 \pm 0.004$ | $0.758 \pm 0.004$ | $0.976 \pm 0.001$ |
| 9 | GAVI250 | $0.025 \pm 0.001$ | $0.260 \pm 0.004$ | $0.763 \pm 0.004$ | $0.977 \pm 0.001$ |
| 10 | GAVI500 | $0.019 \pm 0.001$ | $0.245 \pm 0.004$ | $0.758 \pm 0.004$ | $0.977 \pm 0.001$ |
| 11 | GAVI750 | $0.014 \pm 0.001$ | $0.227 \pm 0.004$ | $0.744 \pm 0.004$ | $0.975 \pm 0.001$ |
| 12 | Bonferroni | $0.008 \pm 0.001$ | $0.067 \pm 0.002$ | $0.440 \pm 0.005$ | $0.877 \pm 0.003$ |
| 13 | LanDeMetsOBF | $0.054 \pm 0.002$ | $0.466 \pm 0.005$ | $0.927 \pm 0.002$ | $0.999 \pm 0.000$ |
| 14 | SeqC2ST_QDA | $0.022 \pm 0.001$ | $0.037 \pm 0.002$ | $0.092 \pm 0.003$ | $0.220 \pm 0.004$ |

Figure 1: Power Curves (under non-normal increment distributions)

In Section D we additionally report sample (or, equivalently, time) savings that each of the methods produced on average (due to the early effect detection in an interim check).

## B.2  Non-Normal Data

The derivation of YEAST involves the application of the Central Limit Theorem to sums of $X_i$ (increments in the metric difference between the two groups). For a fixed sample size, the quality of the normal approximation depends on the distribution of $X_i$. In this section, we explore the performance of YEAST in situations where the distribution of $Y_i^c$ and $Y_i^t$ is not normal. Specifically, we used two alternative distributions: Student's t which has heavier tales than the normal distribution and Gamma which is asymmetric. Student's t distribution had 3 degrees of freedom and was shifted by $\sqrt{3}$ for the control and by $\sqrt{3}(1+\xi)$ for the treatment. The shifting was done to maintain the same coefficient of variation as in Section B.1). The Gamma distribution had its shape parameter set to 1.0. The scale parameter equaled 2 for the control and $2(1+\xi)$ for the treatment. The effect size $\xi$ took the same values as in the first simulation experiment: $\xi = 0.0, 0.1, 0.2, 0.3, 0.4$. The random seed was set to 2023 for the simulations with the Student's t distribution and to 2024 for the simulations with the Gamma distribution. The two seeds were different to avoid dependence across the two sets of simulations.

The results are depicted in Figure 1. The first data point on each line corresponds to the case of no treatment effect and therefore represets the false detection rate. The remaining points represent the power for different treatment effect sizes. Similarly to the experiment with normal data, YEAST had a considerably higher power curve (both for the Gamma and Student's t distribution cases) than other methods that did not inflate the false detection rate.

## C  Discrete Monitoring

In this section, we report additional evaluation results for the case where YEAST was employed in the discrete mode (i.e., with only a fixed number of interim checks). The evaluation was performed against the same replications as in Section B.1. We again follow the protocol from [28] and perform 14, 28, 42, and 56 significance checks (spaced equally across the timeline). Since in these evaluations, we operate in a discrete setting, we were able to include the discrete baselines from [28] in the comparison. Namely, we compare against the following benchmark.

**GST** The group sequential test with alpha spending as proposed in [8]. The test performs a significance check after the arrival of each batch of observations. To schedule a prespecified number of checks it therefore needs an estimate of the total number of observations that would be collected during the experiment time frame. In the evaluations, we consider three different scenarios: when the sample size is estimated precisely (right number of checks), when the sample size is underestimated (leading to more checks than planned), and when the sample size is overstimated (leading

to making fewer checks than planned). The respective entries in the evaluation table are referred to as *GST*, *GSToversampled*, and *GSTundersampled*, respectively. The actual sample size was 500 and the respective sample size estimates for the three scenarios were 500, 250, and 750. In the case of oversampling (i.e., the sample size is underestimated), we apply the correction to the bounds proposed in [37, pp. 78–79]. We consider quadratic and cubic alpha-spending, the latter having the "phi3" prefix in the name.

We report the share of replications where the test detects an effect (i.e., significance is detected in at least one of the interim checks). Table 5 ("False Positive Rate") reports this share for the case where no actual effect was present. All the compared methods except oversampled versions of GST keep the false detection rate below the nominal level of 5%. Table 6 ("Power") contains the share of replications with a detection for the case where the treatment effect was set to 0.2 standard deviations. Methods with an inflated false detection rate were excluded from the power comparisons. The GST method with cubic alpha-spending demonstrated the highest power, closely followed by YEAST and GST with quadratic alpha-spending. We find it remarkable that our proposed method, despite supporting continuous monitoring, performed on par with the GST method in the discrete monitoring setting.

Table 5: False Positive Rate

|   | type | 14 | 28 | 42 | 56 |
|---|------|-----|-----|-----|-----|
| 1 | YEAST | 0.04 | 0.04 | 0.04 | 0.04 |
| 2 | GST | 0.05 | 0.05 | 0.05 | 0.05 |
| 3 | GSTphi3 | 0.05 | 0.05 | 0.05 | 0.05 |
| 4 | GSToversampled | 0.07 | 0.07 | 0.07 | 0.08 |
| 5 | GSToversampledphi3 | 0.09 | 0.10 | 0.10 | 0.07 |
| 6 | GSTundersampled | 0.03 | 0.02 | 0.03 | 0.03 |
| 7 | GSTundersampledphi3 | 0.01 | 0.01 | 0.01 | 0.01 |

Table 6: Power (under a treatment effect of 0.2 standard deviations)

|   | type | 14 | 28 | 42 | 56 | stream |
|---|------|-----|-----|-----|-----|--------|
| 1 | YEAST | 0.90 | 0.91 | 0.91 | 0.91 | 0.92 |
| 2 | GST | 0.90 | 0.90 | 0.90 | 0.89 | - |
| 3 | GSTphi3 | 0.93 | 0.92 | 0.93 | 0.93 | - |
| 4 | GSTundersampled | 0.83 | 0.82 | 0.82 | 0.82 | - |

# D   Sample/Time Savings

The main benefit of sequential testing is the ability to stop the experiment early (once significance is flagged upon one of the interim checks). This allows saving time and, if the treatment is harmful, reduce the negative impact. Thus, the amount of savings that is generated by a sequential test on average is another important metric and we report the savings observed in the experiment from Section B.1 in Table 7. The savings are measured as follows: if a method identified the effect after the arrival of 10% of the total number of observations, the associated sample (or, equivalently, time) savings would be 90%.

Table 7: Simulation Experiment: Sample/Time Savings, %

| | effect size | 0.1 | 0.2 | 0.3 | 0.4 |
|---|---|---|---|---|---|
| | method | | | | |
| 1 | YEAST | 13 | 39 | 58 | 69 |
| 2 | mSPRTphi100 | 9 | 35 | 62 | 75 |
| 3 | mSPRTphi11 | 12 | 41 | 68 | 82 |
| 4 | mSPRTphi25 | 12 | 41 | 68 | 81 |
| 5 | GAVI250 | 12 | 40 | 67 | 80 |
| 6 | GAVI500 | 10 | 37 | 63 | 77 |
| 7 | GAVI750 | 8 | 34 | 60 | 74 |
| 8 | Bonferroni | 2 | 19 | 45 | 67 |
| 9 | LanDeMetsOBF | 14 | 41 | 60 | 70 |
| 10 | SeqC2ST-QDA | 3 | 6 | 14 | 27 |

