# OpenReview forum: "YEAST: Yet Another Sequential Test"
_NeurIPS.cc/2025/Conference — NeurIPS 2025 poster_

### Official Review · Reviewer_n3tA · 2025-06-08

**Clarity:** 2
**Significance:** 2
**Originality:** 2
**Rating:** 4
**Confidence:** 3

**Summary:**

This paper introduces a sequential test for online A/B testing. The method allows unlimited interim checks without inflating type-I error by tracking the cumulative difference in metrics between control and treatment groups against a constant boundary derived by holdout data. The key innovation is deriving this boundary by "inverting" bounds from generalized Levy inequalities. The authors validate their approach using semi-synthetic experiments on real-world "Online Retail" dataset, demonstrating superior Type-I error control and statistical power compared to existing methods like mSPRT and GAVI.

**Questions:**

Please improve Sections 1-2 as mentioned in the weaknesses.

Q1: Robustness to model misspecification: How sensitive is YEAST to violations of its assumptions, particularly when the treatment affects event frequency or outcome variance? Can the authors provide theoretical or empirical analysis of performance degradation in such scenarios?

Q2: Comparison with martingale based and group sequential tests: Why were GST and martingale based methods excluded from the empirical comparison? Those methods are widely used and provide an important benchmark, specifically methods that build on coin-betting.

Q3: Ablation study: What is the impact of N on the test? It’s crucial to assess the impact of the sample size on the test.

**Ethical Concerns:**

["NO or VERY MINOR ethics concerns only"]

**Final Justification:**

This is a borderline paper. I decided to update my final rating to 4 (weak accept). The authors did address my major concern about the comparison to existing work, which I appreciate, although it remains difficult to know exactly how the comparison was implemented.

I am still concerned about the assumption that the frequency of events in each group is the same (Eq. 4): it is quite a strong assumption. While the authors explained their reasoning for making this assumption and vaguely suggested possible relaxations, clear guidelines on how to address this issue in practice would be important.

**Limitations:**

The authors adequately acknowledge key limitations in Section 5, particularly the restrictive null hypothesis and potential oversensitivity to variance changes. However, the discussion could be expanded to include more detailed analysis of when the assumptions are likely to be violated in practice and mitigation strategies beyond those briefly mentioned.

**Paper Formatting Concerns:**

Referneces should be improved.

**Quality:**

2

**Strengths And Weaknesses:**

**Strengths**

S1. Theoretical foundation: The paper presents a rigorous approach by inverting bounds from generalized Levy inequalities (with several weaknesses, as will be detailed below). The theoretical results (Theorems 1 and 2) provide formal guarantees for asymptotic false detection rate control and asymptotic power analysis.

S2. Practical simplicity with a concrete guideline for parameter tuning: YEAST requires estimating only two parameters ($N$ and $V_N$) that can be reliably obtained from pre-experiment data using standard statistical techniques.

S3. Important practical insight about variance estimation: The paper explains the effect of the error in variance estimation.

S4. Superior performance compared to tested benchmarks: While all methods tested control the type-I error rate, YEAST is the only method (out of the ones compared) that obtained power comparable to non-sequential t-tests.

**Weaknesses**

W1 (main issue). Incomplete comparison with group sequential tests and martingale based tests: While GST and martingale methods are mentioned in related work, they are not included in the empirical comparison, making it difficult to assess YEAST's performance against this important class of methods. There is a massive recent progress in using e-processes for online testing, which is somewhat ignored. For example, it is not clear at all why this approach is better than sequential classification-based testing the optimize the coin-betting strategy using ONS?

Podkopaev, Aleksandr, and Aaditya Ramdas. "Sequential predictive two-sample and independence testing." Advances in neural information processing systems (2023).

W2. Asymptotic validity: In contrast to other existing tests, such as GAVI, the proposed test relies on asymptotic assumptions to grant its type-I error control.

W3. Limited scope of null hypothesis:  Assuming that the frequency of events in each group is the same is quite restrictive. Why not relaxing this assumption?

W4. Requirement for pre-specification of N and $V_N$: The method requires fixing the maximum number of events N beforehand, which limits its flexibility compared to truly adaptive methods. While the authors argue this is practical, it may not suit all experimental contexts. Regarding the estimation of $V_N$ - although affecting the validity of the test, I think that using robust estimation is a good way to deal with events that are correlated by nature (such as customers’ actions as presented in the paper).

W5. Limited experimental validation: The evaluation relies on a single dataset and semi-synthetic experiments. Validation on diverse domains would strengthen the claims significantly. Also, an ablation study for different N values can be beneficial.

W6. Clarity and writing quality: The writing quality can be improved, specifically:
- It takes some time to understand what is the exact problem the paper deals with (A/B testing). It will be good to give concrete examples early on.

- It might be understood from the introduction that the paper presents a non-asymptotic valid sequential test, which is not the case.

- The related work section is not comprehensive: In Section 2.1 you detailed about martingale methods without explaining the essence of testing martingales. In Section 2.3 you mentioned that YEAST can be viewed as a generalisation of Miller (2015) test, even without briefly presenting its method. More recent developments in e-process are missing.

- Citations don’t fit the text (either use numerical citations, or adjust the text to fit the named citations). Some references are broken, e.g., in line 101. Please go over the references and improve the citation format.

---

> ### Comment · Reviewer_n3tA · 2025-08-05
>
> The authors did not reply to any of my comments. I maintain my score.

---

> ### Author Response · Authors · 2025-08-05
> **Response to Referee's Comments (Part1: W1-W3)**
>
> Dear Reviewer n3tA,
>
> We would like to thank you very much for the detailed review and apologize for the delay in replying! We were working on implementing an additional baseline that you kindly pointed us at in your review and that delayed our response. Please find our reply below.
>
> **W1:**
> * GST:
>
> (a) originally we have not included GST in the empirical validation section because we viewed it as an approach for discrete monitoring (while our focus has been continuous testing). However, prompted by the reviewer's comments we have since added a continuous version of Lan DeMets (with the O’Brien–Fleming alpha spending function) to the comparison:
>
> | Method         | under H0 | relative effect = 0.05 | relative effect = 0.1 | relative effect = 0.2 |
> |----------------|-----|------|-----|-----|
> | Classical (non-sequential) t-test     | 0.04| 0.12 | 0.25| 0.69|
> | YEAST          | 0.04| 0.12 | 0.25| 0.68|
> | LanDeMetsOBF   | 0.02| 0.07 | 0.16| 0.56|
>
> (b) the supplementary materials we have uploaded to the repository (https://anonymous.4open.science/r/yeast-C15D/additional_evaluation_results.pdf) report our experimental results on synthetic data and have a section where YEAST is deployed in a discrete mode (i.e. when the boundary is set for continuous monitoring but we only do a fixed number of looks) and compared to GST, with the latter showing a slightly better control of the Type-1 error, which is, perhaps, not surprising given that the boundary of YEAST is set for continuous monitoring while the respective GST benchmark specifically expected the fixed of looks.
> * Sec-C-2ST-ONS
>
> Thank you so much for pointing us to a relevant alternative method! In response, we have added its implementation to the repository (https://anonymous.4open.science/r/yeast-C15D/methods/sec_c_2st_ons_qda.R) and are now computing the experimental results (that is why it took us a bit longer to respond, Sec-C-2ST-ONS came out as underpowered in the experiment on the first run hence we are double-checking the implementation). We will share the results including Sec-C-2ST-ONS shortly.
>
> **W2.** Asymptotic validity: In contrast to other existing tests, such as GAVI, the proposed test relies on asymptotic assumptions to grant its type-I error control.
>
> That is a fair point - our core results use CLT approximation. Yet, the empirical results suggest that the method has practical applicability despite the reliance on the asymptotic theory.
>
> **W3.** Limited scope of null hypothesis: Assuming that the frequency of events in each group is the same is quite restrictive. Why not relaxing this assumption?
>
> The difficulty in restricting this assumption lies in the fact that our theory uses Levy’s inequality for bounding the detection rate and that requires the median of  the “residual sum” (or the sum of the remaining increments until the end of monitoring) to be 0 under H0. Intuitively, in the early steps of monitoring, this should not be a problem because the number of remaining increments is large and their sum’s distribution would be approximately symmetric due to the Central Limit Theorem. Yet, closer to the end of monitoring, this deviation can be considerable. Mitigation strategies can include stopping the monitoring early or setting the threshold as if the monitoring is to be run for longer than in reality. The effectiveness of those mitigation strategies and the tradeoff with the power would depend on how quickly the distribution of increment sums converges to a normal distribution. We see this as a large topic for future research.

---

> ### Author Response · Authors · 2025-08-05
> **Response to Referee's Comments (Part 2: W4-W5)**
>
> In the continuation of our previous comment, please find our reply to W4 and W5.
>
> **W4.**
>
> We agree that there are contexts in which the need to pre-specify $N$ and $V_N$ can be limiting (and we would try to make this more explicit in the camera-ready version). Yet,
> * we believe that in a large case of practical situations (particularly in the case of online A/B testing) $N$ and $V_N$ can be reliably estimated from pre-experimental data. (In fact, the commonly used power analysis and experiment sizing procedures involve variance estimation from pre-experimental data and implicitly assume that the respective estimates would be applicable to the experimental period.);
> * the decision boundary of YEAST depends on the square root of N, which makes it more robust to inaccuracies in the estimation of N.
>
> When conducting empirical evaluations on real-world data we have never assumed that N is known and estimated it from pre-experiment data (as part of the method). Hence the empirical results incorporate the uncertainty in the estimation of N. (The estimation was done by simply setting N to the number of events collected during a pre-experiment period of the same length as the time duration of the experiment.)
>
> Similarly to the response to another review, we would like to mention that the simplicity of YEAST makes it possible to quantify analytically how its FDR control would respond to inaccuracies in the plugged N. Specifically, if we overestimate or underestimate $N$ by a factor of $(1+r)$ the effective threshold used during monitoring will equal $z_{1−\alpha/2} \sqrt{1+r}\sqrt{N V_N}$ and the false detection rate will be (approximately) bound by $2 (1 − \Phi(z_{1−\alpha/2} \sqrt{1+r})$ instead of $\alpha$. For example, if $N$ is overestimated by 20% the FDR of YEAST will be bounded by 3% instead of the nominal 5% and if $N$ is underestimated by 20% the FDR will be bounded by 8% instead of 5%.
>
> **W5.**
>
> (a) While working on YEAST, we have additionally validated it in experiments on synthetic data and have now shared their results in the repository (https://anonymous.4open.science/r/yeast-C15D/additional_evaluation_results.pdf). In Section 1, they include variants of YEAST with under/overestimated N (i.e. an ablation study that you mentioned).
>
> (b) Please note than in the experiment on real-world data that is reported in the main text of the manuscript N was estimated from the data (as part of the method) hence the results are reflective of the uncertainty around N.
>
> (c) We had to resort to semi-synthetic experiments as there doesn’t exist (as far as we are aware) any open large-enough datasets with the data of real experiments that we could use to assess the power of the tests.
>
> (d) We have additionally assessed the Type-I error of YEAST on an internal proprietary data set with similar conclusions as for Online Retail. Overall, we didn’t see adding experiments on more data sets particularly strengthening as one can always argue that all of them were cherry-picked regardless of how many we use. But we will strive to produce additional results before the end of the discussion period. Please let us know if you would like us to use some specific data set that is of a particular interest. Unfortunately, there does not seem to be any standard ones for evaluating sequential tests.

---

> > ### Comment · Reviewer_n3tA · 2025-08-07
> >
> > I thank the authors for the detailed reply and efforts in conducting new experiments as well as clarifying the issues I've raised.

---

> > > ### Author Response · Authors · 2025-08-09
> > >
> > > We have checked and corrected the SeqC2ST implementation (using QDA as the classification algorithm). In the semi-synthetic experiment SeqC2ST_QDA shows overrejection likely caused by the dependence between observations (the violation of the i.i.d. assumption): when we replace the actual (non i.i.d.) order values with simulated i.i.d. values the rejection rate of SeqC2ST_QDA becomes closer to the nominal level:
> > >
> > > Type-I Error
> > > | Method                                       | w/ actual (non-iid) order values | w/ simulated (iid) order values |
> > > |----------------------------------------------|-----------------------------------|----------------------------------|
> > > | YEAST (w/ robust variance estimator)         | 0.046                             | 0.049                            |
> > > | LanDeMetsOBF (w/ robust variance estimator)  | 0.023                             | 0.018                            |
> > > | SeqC2ST_QDA                                  | 0.118                             | 0.044                            |
> > >
> > >
> > > At the same time, YEAST and LanDeMetsOBF demonstrate robustness (when the robust variance estimator is used). This can be considered as a downside of SeqC2ST_QDA (and e-values in general) as they rely on the i.i.d. assumption (see e.g. Definition 1 in Podkopaev, Aleksandr, and Aaditya Ramdas) while YEAST (equipped with a robust variance estimator) can handle the non-i.i.d. case. Note that the non-i.i.d. case arises naturally, e.g. in the example of revenue monitoring the individual incoming observations (orders) are not i.i.d. (as some of them are placed by the same users).
> > >
> > > Thank you again for pointing us at a relevant additional benchmark! We will make sure to include it in the manuscript.

---

### Official Review · Reviewer_aybv · 2025-06-13

**Clarity:** 3
**Significance:** 2
**Originality:** 2
**Rating:** 2
**Confidence:** 3

**Summary:**

This paper addresses sequential statistical tests for randomized online experiments. In these experiments, we focus on a specific metric of interest, comparing a control group and a treatment group. Our objective is to continuously track this metric and determine if, at any point, the observed difference between the groups is sufficiently large to conclude that the treatment has had an impact on the metric.

For this setting, the authors propose a novel testing procedure. This method involves tracking the running sum $S_n$​, which represents the difference in the total value of the metric of interest between the treatment and control groups. If this difference crosses a predefined constant alerting boundary, it is concluded that the treatment has an impact on the metric. The paper demonstrates that this method effectively controls the false detection rate and that, when a treatment effect exists, it can be detected with high certainty given sufficiently long monitoring. The effectiveness of this proposed method is further validated through semi-synthetic simulation experiments using real-world data.

**Questions:**

N/A

**Ethical Concerns:**

["NO or VERY MINOR ethics concerns only"]

**Final Justification:**

The authors have addressed many of my concerns, but I am still not sure whether the results are significant.

**Limitations:**

yes

**Quality:**

3

**Strengths And Weaknesses:**

On the positive side, the presentation of the paper is clear and well-structured, making the proposed method and its derivation accessible. Additionally, the empirical results presented in Section 4 show that YEAST achieves accurate false detection rate control and high power.


On the negative side, while Theorems 1 and 2 provide theoretical guarantees for a sufficiently large monitoring period N, the paper does not determine what constitutes a sufficiently large N in real-world scenarios. This lack of clear guidance regarding the practical choice of N limits the theoretical depth of the proposed method, as its performance guarantees are asymptotic. It would be valuable to explore how the choice of N impacts the test's performance.


Furthermore, the results of the following paper would be applicable to the setting:
> Oufkir, A., Fawzi, O., Flammarion, N. and Garivier, A., 2021. Sequential algorithms for testing closeness of distributions. Advances in Neural Information Processing Systems, 34, pp.11655-11664.

Minor comment:
- l.101: "Section ??"

---

> ### Author Response · Authors · 2025-08-05
>
> Dear Reviewer aybv,
>
> Thank you for reviewing the paper!
>
> * As our theory uses the CLT asymptotics, the usual guidance/practice regarding the sample size is applicable. In particular, this is usually not a concern in the context of online A/B testing where sample sizes are large and the distributions are relatively well-behaved for CLT approximations to work well. This is further supported for our empirical assessment in which even on a relatively small-scale data (Online Retail), YEAST demonstrated effective control of Type-I error (and was able to maintain the power at the level of the standard (non-sequential) t-test).
>
> * As is explained in the paper, we advise to estimate $N$ from pre-experimental data (and this is how the method was implemented in production and for the experiment on semi-synthetic data). We have also uploaded the results of additional experiments on synthetic data (https://anonymous.4open.science/r/yeast-C15D/additional_evaluation_results.pdf) which include the assessment of how the choice of $N$ affects the method's performance (please see Section 1 in there). We would also like to mention that the simplicity of YEAST makes it possible to quantify analytically how its FDR control would respond to inaccuracies in the plugged N. Specifically, if we overestimate or underestimate $N$ by a factor of $(1+r)$ the effective threshold used during monitoring will equal $z_{1−\alpha/2} \sqrt{1+r}\sqrt{N V_N}$ and the false detection rate will be (approximately) bound by $2 (1 − \Phi(z_{1−\alpha/2} \sqrt{1+r})$ instead of $\alpha$. For example, if $N$ is overestimated by 20% the FDR of YEAST will be bounded by 3% instead of the nominal 5% and if $N$ is underestimated by 20% the FDR will be bounded by 8% instead of 5%.
>
> * Regarding the paper you have mentioned (thank you for bringing this up) - to the best of our understanding it is applicable to discrete distributions whilst we have been focusing on devising and comparing against universal sequential approaches (suitable for any type of metrics). That said we would mention the paper in the literature section in the camera ready version for completeness.
>
> Please let us know if this addresses your concerns. We would be happy to answer any follow-up questions you might have.

---

> > ### Comment · Reviewer_aybv · 2025-08-06
> >
> > Thank you for your detailed response.
> > However, I still find it unclear how Theorems 1 and 2 should be used in practice for choosing an appropriate value of N. Also, since there are many existing studies on sequential testing, I am not yet sure whether relevant results from other research could be borrowed or adapted here.

---

> > > ### Author Response · Authors · 2025-08-06
> > >
> > > Thank you for the quick reply and suggestions!
> > > We fully agree that analyzing the behavior of YEAST in finite samples (possibly using more advanced analysis techniques) is a very interesting topic for further research but we would rather treat it as out of scope of this paper. We believe that the asymptotic results (Theorems 1 and 2) provide sufficient justification of the method for practical purposes. Think of the classical t-test, for example - being the work-horse of modern A/B testing it is applied routinely to non-normal samples - even though, to the best of our understanding, it is only justified from the asymptotic (CLT) perspective in such scenarios. So our point is that (at least in the important case of online A/B testing) CLT works most of the time and there is no need to devise specific guidance to make sure the sample size is "large enough".

---

### Official Review · Reviewer_aBPe · 2025-06-24

**Clarity:** 3
**Significance:** 3
**Originality:** 1
**Rating:** 5
**Confidence:** 3

**Summary:**

The authors propose a new sequential hypothesis testing method called YEAST. It generalizes a previous method for count-type metrics to any type of metric. Compared with alternative methods, YEAST offers more power by better controlling the Type-I error rate. This claim is supported by semi-synthetic experiments based on real-world data. The paper also provided theoretical guarantees for the error control and power of YEAST.

**Questions:**

1. How is YEAST different from alpha spending with the O’Brien–Fleming function?
2. Does YEAST stop significantly earlier than N (both under H_0 and H_1)?

**Ethical Concerns:**

["NO or VERY MINOR ethics concerns only"]

**Final Justification:**

Authors addressed both my concerns adequately in their response.

**Limitations:**

The authors have a substantial section discussing the limitations of their method (more detailed than a typical submission).

**Paper Formatting Concerns:**

Citations that need to be parenthetical are written in in-text format, for instance Wald (1947) instead of (Wald, 1947).

**Quality:**

3

**Strengths And Weaknesses:**

The paper is written very clearly and the exposition is very easy to follow. In particular, I really appreciated the informal introduction of the method in (6) and (7) before the formal proof in Theorem 1.

The method requires estimating V_N (the normalized variation of the metric being monitored). This is a limiting factor of the method but the authors have a nice discussion regarding the impact of the estimation error on the Type-I error based on their Theorem 1.

My main concern is the novelty of the proposed method. Is it not the case that checking $S\_n>b^*=z\_{1-\alpha/2}\sqrt{N\hat{V}\_N}$ is essentially an alpha spending strategy? More precisely, the false discovery rate until the $n$-th sample can be bounded as

$$P(\max \_{i=1}^n S\_i > b^* ) \leq 2 P(S\_n > b^*) \leq 2(1-\phi(z\_{1-\alpha/2}/\sqrt{n/N}))$$

which is exactly the O’Brien–Fleming alpha spending function. It is not clear to me what is the contribution of the paper to this strategy (I may have missed an important nuance). Because of this concern, I have rated the paper a borderline reject. However, if the authors highlight the novel parts of their approach during the rebuttal, I am open to increasing my score as I like the simplicity of the paper otherwise.

Related to my previous point, the baselines in the experiment section do not seem to include any alpha spending strategy.

Experiments show that YEAST has comparable error control and power to a non-sequential test. However, the authors do not consider stopping time as a metric. Does YEAST use almost all of the N samples to be able to achieve this comparable performance? If the answer is yes, its benefit over a non-sequential test would be very limited.

---

> ### Author Response · Authors · 2025-08-03
> **Response to questions and weaknesses**
>
> Dear Reviewer aBPe,
>
> Many thanks for reviewing the paper! Please find our response to your questions below.
>
> 1. YEAST and OBF are fundamentally different as OBF uses a statistic normalized by the observed information (i.e., the variance of the cumulative data up to time n), and adjusts the rejection threshold at each look. In contrast, our method normalizes the partial cumulative sum $\sum_{i=1}^n X_i$​ by the **variance of the full dataset** ($var(\sum_{i=1}^N X_i)$) and compares it to a **fixed threshold**. As a result, the boundary of YEAST is constant (does not change from one look to another) while this is not the case for OBF.
> When preparing the manuscript we considered alpha-spending as a discrete monitoring method and left it out for that reason but prompted by your question we have now included a continuous version of the alpha-spending approach (Lan DeMets with the OBF spending rule) in the validation experiment and reran it. We have uploaded the implementation to the repository (see [here](https://anonymous.4open.science/r/yeast-C15D/methods/ld_obf.R)) and updated the [experiment script](https://anonymous.4open.science/r/yeast-C15D/evaluate_on_online_retail.R). The detection rates are reported in the table below. YEAST outperformed the OBF alpha-spending approach in the experiment.
> | Method         | under H0 | relative effect = 0.05 | relative effect = 0.1 | relative effect = 0.2 |
> |----------------|-----|------|-----|-----|
> | Classical (non-sequential) t-test     | 0.04| 0.12 | 0.25| 0.69|
> | YEAST          | 0.04| 0.12 | 0.25| 0.68|
> | LanDeMetsOBF   | 0.02| 0.07 | 0.16| 0.56|
>
> This difference in the result additionally highlights that YEAST and OBF alpha-spending are two different methods and that YEAST is novel.
>
> 2. That is a valid point! We have actually conducted an assessment of when the stopping occurs and can confirm that YEAST does stop earlier than N. (We have not included the sample "savings" results in the main text due to space limitations but would add them to the camera-ready version given the point you have raised.) Some time savings results can be found in Section 3 of the supplementary materials we have now uploaded to the repository (https://anonymous.4open.science/r/yeast-C15D/additional_evaluation_results.pdf). Those pertain to the experiment on synthetic data. We will compile the time savings results for the main experiment (i.e. the one on the Online Retail dataset) and share them here shortly. As a side-comment, we have implemented YEAST and used it in production for monitoring real-world A/B tests where it has led to early detection of regressions and early stopping on multiple occasions.

---

> > ### Comment · Reviewer_aBPe · 2025-08-06
> >
> > Thank you for the response, it addresses both my concerns, and I have increased my score as a results. Including results for OBF and reporting stopping times would be both great improvements.

---

> > > ### Author Response · Authors · 2025-08-07
> > >
> > > Thank you again for reviewing the paper and for helpful suggestions! We will make sure to add OBF and stopping times to the manuscript.
> > >
> > > P.S. As a further follow-up on Q2 we have also computed the time savings of YEAST in the main (semi-synthetic) experiment:
> > >
> > > |                               | relative effect = 0.05 | relative effect = 0.10 | relative effect = 0.20 |
> > > | ------------------------------------ | ---- | ---- | ---- |
> > > | YEAST (across all replications)             | 3%   | 7%   | 23%  |
> > > | YEAST (across replications with detections) | 27%  | 29%  | 34%  |

---

### Official Review · Reviewer_eKju · 2025-06-30

**Clarity:** 4
**Significance:** 3
**Originality:** 2
**Rating:** 5
**Confidence:** 4

**Summary:**

A new sequential test procedure is proposed and analyzed. It is experimentally shown to outperform current/recent methods in both false detection rate (FDR) control as well as power. In terms of theoretical analysis, the authors essentially leverage Levy's inequality and the Central Limit Theorem to establish tunable FDR control and that the probability of correctly making a true detection tends to 1 asymptotically.

**Questions:**

**Question 1:**  Regarding Section 4.2's experiments to assess Power, I thought it was just a little curious that to synthetically generate a difference between control and treatment, a multiplier effect was chosen, rather than, say, an additive effect. I think an additive separation between control and treatment signals could be quite a reasonable phenomenon in a variety of contexts. While the results in Table 3 indicate that YEAST can outperform other methods by orders of magnitude under their choice of a multiplier effect, I just can't help but wonder how things behave under the additive effect. And considering the authors have well-documented their code, I wonder if it would be possible to ask for them to reconfigure their experiment to show how YEAST performs (relative to other methods) under the additive effect. On a related note, I am wondering if the multiplier effect setting benefits YEAST in some way that is related to its use of the scaled variance $V_N$ = var($S_N$)/N (or its estimate).

**Ethical Concerns:**

["NO or VERY MINOR ethics concerns only"]

**Final Justification:**

To me, there aren't really any glaring downsides/weaknesses. The authors' efforts to provide follow-up discussions and experiments were satisfactory.

Their Table 3 results indicating much higher power over other methods is intriguing, and they were able to show a similar table under additive effects too.

I personally think this paper gets a 4 at the very least, which was my original score. After the rebuttal, in which everything in my review was addressed, I am willing to give a slight bump to 5.

**Limitations:**

yes

**Paper Formatting Concerns:**

- Throughout the Appendix, there are equation lines that involve duplicated "=" symbols.
- On line 401, if I'm not mistaken, $N\mu$ should in fact be $m_N$.

**Quality:**

3

**Strengths And Weaknesses:**

**Strengths:**
- The idea is short and clear/simple
- YEAST has the capacity to significantly outperform current/recent methods in some settings. Meanwhile, it is not expensive in terms of parameters
- Sequential Testing is a topic that is current

**Negatives:**
- YEAST requires a finite horizon N, whereas another method like GAVI makes no assumptions on the stopping rule. This requirement by YEAST makes it so that it's less straightforward to claim that YEAST is superior to GAVI.
- The semi-synthetic simulations presented haven't fully convinced me that YEAST is necessarily a superior sequential test method (see questions) in terms of FDR control and Power, but I remain open, and look forward to hearing from the authors during the rebuttal period.

---

> ### Author Response · Authors · 2025-08-02
> **Response to Q1**
>
> Dear Reviewer eKju,
>
> Thanks a lot for carefully reading the manuscript! Please find our response to Q1 below:
>
> We used a relative effect in the power study because it makes the results less dependent on the distribution of $Y$ (order values in the example of Online Retail) and hence more generalizable. We will next elaborate why this is the case.
>
> The power of YEAST (just as the power of other tests under comparison) depends on the ratio between the squared mean and variance of the increments (i.e. the signal to noise ratio $E[X_i]^2 / V_N$). The term $V_N$ replaces $var(X_i)$ in our derivations since we allow for dependence between increments. Should they be independent it would reduce exactly to $var(X_i)$: $V_N = var(S_N)/N = var\left(\sum_{n=1}^N X_n\right)/N = N var(X_i)/N = var(X_i)$.
>
> In the additive case ($Y_t = Y_c + \Delta_a$), we have that
> $$
> SNR = E[X_i]^2/var(X_i) = \Delta_a^2 / (4 var(Y_c) + (2 E[Y_c] + \Delta_a)^2)
> $$
> and in the multiplicative case ($Y_t = Y_c (1 + \Delta_r)$) it equals
> $$
> 	SNR = E[X_i]^2/var(X_i) = E[Y_c]^2\Delta_r^2 / (2 var(Y_c) (1 + (1 +\Delta_r)^2) + E[Y_c]^2(2 + \Delta_r)^2).
> $$
> When the treatment effect is absolute, the signal-to-noise ratio depends more strongly on the mean and variance of the baseline outcome $Y_c$​ because the signal - the squared effect size - remains constant, while the noise grows with both the variance and the square of the mean. As a result, increases in $E[Y_c]$​ or $var(Y_c)$ significantly degrade SNR.
> In contrast, when the effect is relative (i.e., proportional to $Y_c$​), both the signal and parts of the noise scale quadratically with $\mu_c$​. This proportional scaling cancels out in the SNR expression, making SNR less sensitive to changes in the cumulants of $Y_c$.
> Therefore, SNR under a relative effect model is more stable across different populations or settings, while SNR under an absolute effect model is more sensitive to shifts in the distribution of the control outcome and hence experiment results using absolute effects are less generalisable.
> That said, it is indeed straightforward to rerun the experiments with an absolute effect. Here are the results for the absolute effect of 15, 35, and 70 pounds, respectively:
> | # | Method    | 15               | 35               | 70               |
> |----|-----------|------------------|------------------|------------------|
> | 1  | Classical | 0.1014 ± 0.0033  | 0.2453 ± 0.0047  | 0.6422 ± 0.0052  |
> | 2  | YEAST     | 0.1016 ± 0.0033  | 0.2424 ± 0.0047  | 0.6337 ± 0.0052  |
> | 3  | GAVI250   | 0.0030 ± 0.0006  | 0.0135 ± 0.0013  | 0.1068 ± 0.0034  |
> | 4  | GAVI500   | 0.0043 ± 0.0007  | 0.0177 ± 0.0014  | 0.1291 ± 0.0036  |
> | 5  | GAVI750   | 0.0057 ± 0.0008  | 0.0211 ± 0.0016  | 0.1435 ± 0.0038  |
> | 6  | mSPRT100  | 0.0052 ± 0.0008  | 0.0195 ± 0.0015  | 0.1374 ± 0.0037  |
> | 7  | mSPRT011  | 0.0014 ± 0.0004  | 0.0076 ± 0.0009  | 0.0728 ± 0.0028  |
> | 8  | mSPRT025  | 0.0023 ± 0.0005  | 0.0109 ± 0.0011  | 0.0935 ± 0.0032  |
> | 9  | GAVI250   | 0.0030 ± 0.0006  | 0.0135 ± 0.0013  | 0.1068 ± 0.0034  |
> | 10 | GAVI500   | 0.0043 ± 0.0007  | 0.0177 ± 0.0014  | 0.1291 ± 0.0036  |
> | 11 | GAVI750   | 0.0057 ± 0.0008  | 0.0211 ± 0.0016  | 0.1435 ± 0.0038  |
> | 12 | GAVI10K   | 0.0107 ± 0.0011  | 0.0392 ± 0.0021  | 0.2200 ± 0.0045  |
>
> Note that the Online Retail is a relatively small data set and hence the sensitivity of all tests is relatively low. Therefore the main result here is that YEAST performs on par with the non-sequential t-test while other sequential methods demonstrate a considerably lower power.
>
> Please let us know if this answers your question and we will be happy to elaborate further.

---

> > ### Author Response · Authors · 2025-08-02
> > **Response to the outlined weaknesses**
> >
> > We would also like to respond to the weaknesses of the paper you have outlined in your review.
> >
> > 1. It is true that YEAST requires setting a finite horizon while GAVI doesn’t. Yet, in practical scenarios we have not found this problematic for the following two reasons:
> > when running A/B experiments it is typical to set a certain duration and the number of observations collected over this time frame in our production setting is quite predictable (and we believe this is a common case in the industry),
> > the decision boundary of YEAST depends on the square root of N, which makes it more robust to inaccuracies in the estimation of N.
> > When conducting empirical evaluations on real-world data we have never assumed that N is known and estimated it from pre-experiment data (as part of the method). Hence the empirical results incorporate the uncertainty in the estimation of N. (The estimation was done by simply setting N to the number of events collected during a pre-experiment period of the same length as the time duration of the experiment.)
> > That said, we fully acknowledge that there are situations where it is difficult to reliably estimate N and it is indeed better to resort to methods like GAVI that do not require setting a finite horizon. Prompted by your review, we would elaborate on this in the camera-ready version of the paper. More generally, we believe that “one method fits all” scenarios are very rare in reality and would like YEAST to complement the existing sequential testing toolbox rather than replace any of the existing methods fully.
> > As a side-note, we would like to mention that the simplicity of YEAST makes it possible to quantify analytically how its FDR control would respond to inaccuracies in the plugged N. Specifically, if we overestimate or underestimate $N$ by a factor of $(1+r)$ the effective threshold used during monitoring will equal $z_{1−\alpha/2} \sqrt{1+r}\sqrt{N V_N}$ and the false detection rate will be (approximately) bound by $2 (1 − \Phi(z_{1−\alpha/2} \sqrt{1+r}))$ instead of $\alpha$. For example, if $N$ is overestimated by 20% the FDR of YEAST will be bounded by 3% instead of the nominal 5% and if $N$ is underestimated by 20% the FDR will be bounded by 8% instead of 5%.
> >
> > 2. To provide further empirical evidence we have uploaded additional evaluation results to the paper repository (https://anonymous.4open.science/r/yeast-C15D/additional_evaluation_results.pdf). These use synthetic data and were not added to the paper due to space limitations.
> >
> > Please let us know if this response has addressed your concerns. We are happy to answer any follow up questions you might have.

---

> > > ### Comment · Reviewer_eKju · 2025-08-05
> > > **On the need for a finite N**
> > >
> > > I thank the authors for their thorough reply on the matter of a finite horizon. I wholeheartedly agree and understand the authors' stance that a “one method fits all” scenario is very rare. Their comments from the practical and theoretical perspective are satisfactory in my opinion.
> > >
> > > Their characterization of the relation between the quality of approximation of N and the FDR control is much appreciated too.
> > >
> > > To be clear, this matter was never a dealbreaker for me per se...rather, a point for comparison with other methods. And on that note, the authors' subsequent comparison with GAVI is appreciated, as are their proposal to add mention of this in a camera-ready.

---

> > ### Comment · Reviewer_eKju · 2025-08-05
> > **Relative versus Additive Effect**
> >
> > I thank the authors for their thorough rebuttal.
> >
> > Their explanation about the sensitivity/stability matters is appreciated, as is the new table with results on additive effect.

---

### Note · Authors · 2025-08-13

We thank the reviewers for their thoughtful feedback, which has helped clarify YEAST’s contributions and applicability.

Novelty and Distinction. As clarified in the rebuttal, YEAST is not equivalent to O’Brien-Fleming alpha spending. It uses a constant boundary and normalizes by the variance of the full horizon, rather than observed information, yielding different statistical properties and superior performance. New experiments with a continuous OBF implementation confirm YEAST’s higher power with FDR control. We also implemented the reviewer-suggested Sec-C-2ST-ONS coin-betting method and found it overrejects under non-i.i.d. conditions, while YEAST with robust variance estimation maintains proper control (4.6%), making it practically superior to e-values for real A/B testing scenarios where customer behaviors create dependencies.

Practicality of Finite Horizon N. While YEAST requires a pre-specified horizon, this is rarely restrictive in online experiments, where durations and sample sizes are routinely estimated for power analysis. Our evaluation already incorporates the uncertainty in N by estimating N from pre-experiment data. Analytically, even a 20% misestimate changes FDR bounds only modestly.

Evaluation Breadth. In addition to the semi-synthetic results in the paper, we've uploaded further synthetic-data and discrete-monitoring comparisons to the repository. They confirm YEAST’s superior Type-I error control and power properties, as well as show it achieves meaningful early stopping. Additive-effect experiments also show YEAST matching t-test performance, unlike competing sequential methods.

Scope and Assumptions. We acknowledge the asymptotic nature of our theoretical guarantees and the strong null hypothesis. However, empirical evidence shows good finite-sample performance, and analytical tractability lets practitioners assess and manage the impact of assumption violations. Pre-specifying N, while limiting in some contexts, aligns well with standard A/B testing practice where experiment durations and expected sample sizes are typically planned in advance.

In sum, YEAST offers:
* a novel constant-boundary derivation,
* strong empirical performance - uniquely matching non-sequential t-test power while controlling FDR,
* robustness in natural non-i.i.d. settings,
* transparency about limitations and their impact.

We believe YEAST meaningfully advances the sequential testing toolbox for real-world experimentation and merits acceptance.

---

### Decision · Program_Chairs · 2025-09-17

**Decision:**

Accept (poster)

**Comment:**

This paper introduces a sequential test for online A/B testing. For the theoretical analysis, the authors use Levy's inequality and the Central Limit Theorem to establish tunable FDR control and that the probability of correctly making a true detection tends to 1 asymptotically.

The initial reviews had somewhat diverging opinions and ratings. All reviewers appreciated the discussion with the authors which allowed to clarify a number of points. One reviewer still maintained their low rating (2) due to "uncertainty about significance of the results".

Overall with 4 positive reviewers (2 strongly so) and that the discussion appeared to be quite constructive and fruitful, I recommend to accept the submission.